# Relative and Cumulative Effects of Climate and Land Use Change on Hydrological Ecosystem Services in Northeast China

**Mengqi Wang** [1,2] and **Guoping Lei** [1,2,*]

1   School of Humanities and Law, Northeastern University, Shenyang 110169, China; 1910006@stu.neu.edu.cn
2   Key Laboratory of Land Protection and Use, Department of Natural Resources of Liaoning Province, Shenyang 110169, China
*   Correspondence: guopinglei@126.com

**Abstract:** Climate change (CC) and land use change (LUC) have been determined as two major environmental change variables that broadly affect hydrological ecosystem services (HESs). However, the relative and cumulative effects of CC and LUC on HES at large spatial scales where there is great environmental heterogeneity is still unclear enough to support the formulation and update of land use decision-making and ecological management policies. This study has quantified the spatiotemporal change of HESs (water yield, water purification, soil retention) from 1992 to 2020 in northeast China, and evaluated the relative contribution and cumulative effects of CC and LUC on HESs through environmental setting scenarios and using two indicators (the Relative Importance Index and the Combined Effects Index). This study yielded the following results: (1) From 1992 to 2020, water yield (WY) (+94.33 mm) and soil retention (SR) ($5.28 \times 10^3$ t/km$^2$) both showed an upward trend from 1992 to 2020 and an upward trend in nitrogen export (NE) indicating a decline in water purification (WP). (2) There was significant spatial heterogeneity of HESs in northeast China, which included significant increases in WY in the Sanjiang Plain; NE in the Songnen Plain (SNP), Sanjiang Plain (SJP), and Liao River Plain (LJP); and SR in the Greater Khingan Mountains (GKMR), Lesser Khingan Mountains (LKMR), and Changbai Mountains (CBMR). (3) WY was more affected by CC than LUC, especially in the SJP, the eastern LRP, and the southern CBMR; NE was more affected by LUC than CC in the western LRP, the southern GKMR, and the southwestern SNP; SR was more affected by LUC than CC in the GKMR; SR was more affected by CC than LUC and intensity gradually increased in the CBMR and LKMR. (4) The cumulative effect of CC and LUC contributed to HESs in most regions but inhibited HESs in some regions; warming and forestland expansion especially significantly inhibited WY. Our study emphasizes that current land use policies and ecosystem management practices should consider the relative and cumulative effects of CC and LUC on HESs to maintain diverse ecosystem services and ensure human well-being.

**Keywords:** hydrological ecosystem services; climate change; land use change; land use decisions; Northeast China

## 1. Introduction

Ecosystem services are the benefits that humans derive directly or indirectly from ecosystems and they play an important role in establishing and maintaining environmental conditions and are the material basis for human survival and development [1]. Global population growth, rapid economic development, major climate change, increased intensity of human activity, and increased demand for natural resources since the 21st century have all contributed to the degradation or unsustainable usage of over 60% of ecosystem services [2,3]. The intensity of degradation of the hydrological ecosystem services (HESs), which include climate regulation, water yield, soil conservation, water purification, and biodiversity, is dramatically increasing at unprecedented levels at global and regional scales [4]. These HESs are the most important drivers for reducing environmental crises,

enhancing human health, and achieving sustainable development goals [5]. Therefore, assessment, management, and conservation of HESs is an important task for policy makers and governments worldwide.

Climate change (CC) and land use change (LUC) have been identified as the most vulnerable elements driving changes in ecosystem services. CC alters ecosystem structure and function through mechanisms that affect biodiversity–function relationships, and directly affects ecosystem services through altering hydrological processes and biogeochemical cycles, such as photosynthesis and greenhouse gas fluxes, nutrient cycling and soil formation, and the timing and volume of water flow [6–8]. Pressure on ES from climate change is exacerbated by increasing urbanization, population growth, and unsustainable expansion [9]. LUC affects ecological processes such as energy exchange and the water cycle by altering the physical properties of the land surface (e.g., evapotranspiration, soil moisture, surface roughness, albedo, and trace gas fluxes) [10,11]. Studies have shown that changes in potential surface properties caused by LUC affect almost all hydrological cycle processes (e.g., precipitation interception, evapotranspiration, and infiltration) and interfere with physical and chemical changes (e.g., interception, absorption, and conversion to non-hazardous pollutants), thereby affecting water quantity, nutrient concentrations, and sediment load [12]. Therefore, clarifying the impacts of CC and LUC on HESs is crucial for policy makers and stakeholders to make effective management decisions based on improved hydrological ecosystem services.

At present, the impact of CC and LUC on hydrological ecosystem services have been widely studied [13,14]. However, their cumulative effect further exacerbates the impact on HESs. Currently, most studies are concentrated on evaluating the impacts of independent CC and LUC on HESs [15,16]. Currently, most studies have focused on assessing the impacts of independent CC and LUC on HESs, considering that an understanding of the mechanisms driving the relative and cumulative effects of CC and LUC is still lacking, and the challenge of establishing mechanisms for ecosystem optimization in response to CC and LUC is enormous. In addition, most studies have focused on administrative [17], watershed [5], typical regional [18], and ecological reserve scales [19], or analyzed the impact of LUCC on HESs as a result of a single decision [20]. Fang et al. studied the impact of climate and land use change on ecosystem services by taking the Yangtze and Yellow River basins as the study area and used a geographically weighted regression model to represent the spatial variation of the drivers [21]. Although the effects of climate and land use change on ecosystem services have been discussed at large spatial scales, studies at large spatial scales have mainly focused on the same type of area, with no significant spatial heterogeneity in topography, vegetation, or climatic characteristics [22]. Lack of assessment of the impacts of CC and LUC on HESs at large spatial scales where there is significant environmental heterogeneity, thereby reducing the regional utility of ecosystem management responses. Therefore, there is an urgent need to study the relative and cumulative effects of LUC and CC on HESs.

The northeast of China is the largest natural forest area and is located in one of the world's four major black soil areas, which occupies an important position in China's food security and ecological safety guarantee system [23]. However, the impact of inter-annual variation in precipitation and the intensity of precipitation increases at a rate of 0.11 mm/day/decade, and total water resources and runoff are also more variable from year to year [24]. Coupled with the impact of global warming, frequent composite climate events such as low-temperature droughts, high-temperature droughts, and other natural disasters such as droughts and floods are frequent, which greatly influences changes in HESs and poses severe challenges to ecological protection, stable food production, and increases in yields [25]. China's national and local governments have promulgated a series of ecological protection and restoration projects to improve multiple ecosystem services, such as the Natural Forest Conservation Project and the National Wetland Conservation Plan [26]. However, with the dramatic increase in the demand for food and economic development, the government has implemented a series of land management policies

involving cultivated land protection, macroeconomic regulatory policies, and regional development strategies, resulting in units of ecological space with important ecological functions being continuously occupied, developed, and destroyed [27,28]. This has profoundly changed land use structure and ecosystem functions, with water, soil, climate, and ecological imbalances occurring in some areas [29]. Therefore, exploring the relative and cumulative effects of CC and LUC with regard to land use decision-making on HESs in northeast China is significant for the formulation and updating of land use management policies and ecosystem restoration projections.

The aim of this study is to provide a comprehensive assessment of HESs in northeast China and to improve understanding of the relative and cumulative effects of CC and LUC driven by multiple land use decisions on HESs in the northeast China. This study aims at the following detailed objectives: (1) revealing the spatiotemporal characteristics of CC and LUC in Northeast China from 1992 to 2020; (2) assessing the spatiotemporal characteristics of HESs (water yield, water purification, soil conservation) in the northeast from 1992 to 2020; (3) establishing four scenarios (two realistic scenarios and two scenarios of climate and land use exchange) to identify the relative and cumulative effects of CC and LUC on HESs. This study can provide key information for ecosystem management and protection, and optimize climate-related policies, strategies, and incentives.

## 2. Materials and Methods

### 2.1. Study Area

Northeast China is located between $115°32'$ E to $135°09'$ E and $38°42'$ N to $53°35'$ N, with a total area of about $1.24 \times 10^6$ km$^2$, covering Heilongjiang, Jilin and Liaoning provinces as well as the eastern part of the Inner Mongolia Autonomous Region. It has a temperate continental monsoon climate zone with large temperature variations throughout the year and annual precipitation ranging from 300–1000 mm. For a long time, northeast China has had various types of natural resources (e.g., forestland, wetland, black soil, etc.), and is fed by major rivers such as the Songhua, Liao, Neng, and Yalu Rivers, which play an important role in food, timber, and mineral resources [27]. High-intensity human activities and large-scale reclamation have led to significant changes in land use patterns, which, together with the complex topographical and geomorphological conditions, have formed a variety of lower-surface characteristics, creating a complex hydrological and ecological process [24,26]. According to the topography, vegetation and climatic characteristics, northeast China can be divided into six geographical regions: the Great Khingan Mountains Region (GKMR), the Lesser Khingan Mountains Region (LKMR), the Changbai Mountains Region (CBMR), the Sanjiang Plain (SJP), the Songnen Plain (SNP), and the Liao River Plain (LRP) (Figure 1).

### 2.2. Quantification of Hydrological Ecosystem Services

The InVEST model has been widely used due to, its advantages of less input data, large export data, visualization of results, and quantitative analysis of abstract ecosystem service functions [30,31]. In this study, the water yield model (for water yield), the sediment delivery ratio model (for soil conservation) and the nutrient delivery ratio model (for nitrogen export) were used to evaluate the corresponding HESs in northeast China.

#### 2.2.1. Water Yield

Water yield (WY) plays a key role in agricultural irrigation and human life. It is the foundation of water resource planning and management [4]. WY was quantified using the water yield model and the InVEST model was used to quantify the WY per pixel based on the Budyko curve with water balance as the core [32]. For each pixel, the annual water yield was calculated by subtracting the actual evapotranspiration from the average precipitation [30]. The following equation was used to calculate water yield in each pixel:

$$Y(x,j) = \left(1 - \frac{AET(x,j)}{P(x)}\right) \cdot P(x) \tag{1}$$

where $Y(x,j)$ (mm) is the water yield for pixel $x$ on the LULC $j$; $AET(x,j)$ (mm) is the actual evaporanspiration for pixel $x$ on the LULC $j$; $P(x)$ (mm) is the precipitation for pixel $x$.

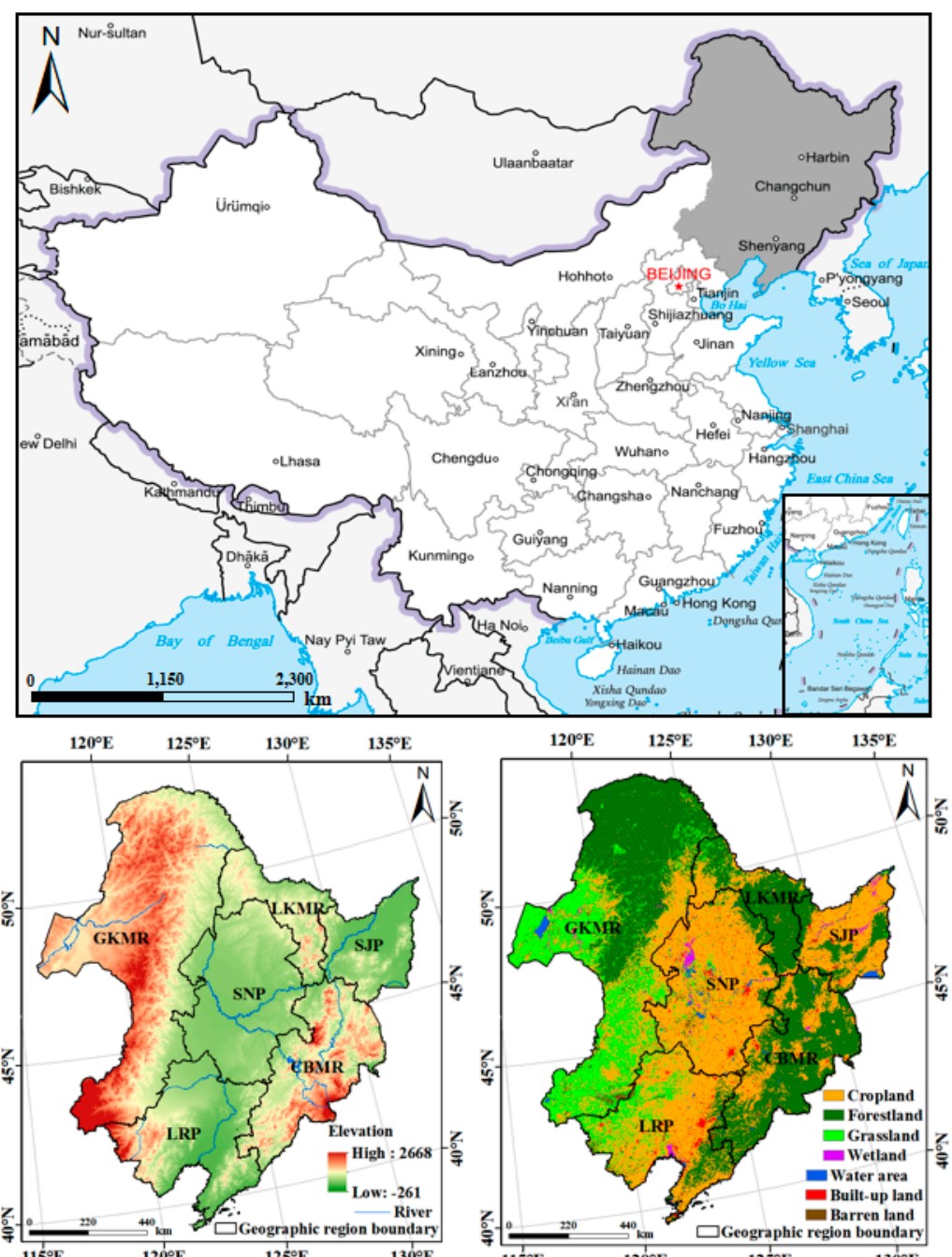

**Figure 1.** Geographic location of northeast China and general geographic features.

### 2.2.2. Water Purification

Since nitrogen and phosphorus are the main pollutants caused by humans, and the data availability of phosphorus is limited, the nitrogen export coefficient is used to represent its water purification capacity [33]. The nutrient delivery ratio model maps nutrient sources from watersheds and nutrient transport to the stream, and uses a simple mass balance method to describe the movement of nutrients in space and then estimates

the nitrogen/phosphorus export of each pixel [28,34]. Higher levels of nutrient exports indicated lower water purification [35]. The following equation was used to calculate nitrogen export in each pixel:

$$ALV_i = HSS_i \cdot pol_i \tag{2}$$

$ALV_i$ is the adjusted load value of pixel $i$. $pol_i$ is the output coefficient of pixel $i$, and $HSS_i$ is the hydrological sensitivity score of the calculation method of pixel $i$:

$$HSS_i = \frac{\gamma_i}{\gamma_w} \tag{3}$$

$\gamma_i$ is the runoff coefficient at pixel $i$, while $\gamma_w$ is the average runoff coefficient index:

$$\gamma_x = log\left(\sum_u \delta_u\right) \tag{4}$$

$\sum_u \delta_u$ represents a spatially varying pixel of runoff potential, referring to the ability to deliver nutrients downstream.

### 2.2.3. Soil Retention

Soil retention is the ability of ecosystems to prevent and mitigate soil erosion [36]. The sediment delivery ratio model used eroded sediment and then computed the SE, which was the soil loss that actually reached the catchment outlet [30]. The equation is shown as follows:

$$SC_i = RKLS_i - USlE_i + UPSD_i \tag{5}$$

where $SC_i$ is the annual soil conservation; $RKLS_i$ is the potential soil loss of bare soil calculated with the USLE formula without considering vegetation cover and water and soil conservation measures; $USlE_i$ is the total potential soil loss; ($RKLS_i - USlE_i$) is the amount of soil erosionreduction; and $UPSD_i$ is the total amount of sediment deposited from upstream sources as a result of retention.

We compared the water yield results simulated by InVEST model with the surface water resources published in the Water Resources Bulletin, and the error was controlled at 0.09, indicating that the simulation was accurate. The results of WP and SR simulated by InVEST model were compared with those of other relevant studies, and the results were consistent [23,26].

### 2.3. Factor Analysis

#### 2.3.1. Land Use and Climate Change Settings

We created four scenarios using two periods of climate and land use data to explore the effects of CC and LUC on HESs in northeast China. Scenario 1, the baseline, was based on real environmental conditions in 1992. Scenario 4 was based on real environmental conditions in 2020. In contrast, in scenario 2, climate was kept constant from 1992 to 2020, leaving land use change as the sole driver affecting changes in ecosystem services. In scenario 3, land cover remained constant from 1992 to 2020, leaving only the effects of climate change to relate to changes in ecosystem services (Table 1). Using this approach, we were able to disaggregate the impacts of different facets of change on ecosystem service provision. Similarly, we could set up change scenarios for 1992–2000, 2000–2010, and 2010–2020.

**Table 1.** Land use and climate scenario settings.

|  | Landuse1992 | Landuse2020 |
| --- | --- | --- |
| Climate1992 | Scenario1 | Scenario2 |
| Climate2020 | Scenario3 | Scenario4 |

### 2.3.2. Relative Importance and Cumulative Effects Analysis

We used the relative importance index (RII) and the combined effect (CEI) to express the relative and cumulative effects of CC and LUC on HESs [36]. A value greater than 0 indicates that LUC has a greater relative importance than CC. A value lower than 0 indicates CC has greater relative importance than LUC, and a value of 0 indicates that the influence of both factors is equal. RII is calculated as:

$$\text{RII} = \frac{|\text{ES}_{\text{scenario2}} - \text{ES}_{\text{scenario1}}| - |\text{ES}_{\text{scenario3}} - \text{ES}_{\text{scenario1}}|}{\max(\text{ES}_{\text{scenario1}})}, \begin{pmatrix} > 0, \text{Land use} \\ = 0, \text{Equal} \\ < 0, \text{Climate} \end{pmatrix} \quad (6)$$

A value greater than 0 indicates land use and climate factors have an synergistic effect on ecosystem service. A value lower than 0 indicates land use and climate factors have a inhibitory effect on ecosystem service. A value of 0 indicates a state independent from the effects of these variables. CEI is calculated as:

$$\text{CEI} = \frac{\text{ES}_{\text{scenario2}} + \text{ES}_{\text{scenario3}} - \text{ES}_{\text{scenario1}} - \text{ES}_{\text{scenario4}}}{\max(\text{ES}_{\text{scenario1}})}, \begin{pmatrix} > 0, \text{Synergistic} \\ = 0, \text{Independant} \\ < 0, \text{Inhibitory} \end{pmatrix} \quad (7)$$

where, $\text{ES}_{\text{scenario1}}$, $\text{ES}_{\text{scenario2}}$, $\text{ES}_{\text{scenario3}}$, $\text{ES}_{\text{scenario4}}$ represents the value of sole or aggregated ecosystem services in each scenario set.

### 2.4. Data Requirement and Preparation

The land use data were provided by the Climate Change Initiative (CCI, http://maps.elie.ucl.ac.be/CCI/viewer (accessed on 21 March 2023)). The relatively high spatial resolution of the data and its long-term consistency, annual updates, and high thematic detail on a global scale make it attractive for numerous applications such as land accounting, forest monitoring, and scientific research [37]. Dominant land use categories agriculture and forest show an agreement of over 80% [38]. Considering the requirement of the study, 37 LC types were reclassified into 7 major land use categories: cropland, forestland, grassland, wetland, water area, built-up land, barren land.

The others data sources was shown in Table 2. The monthly precipitation and evapotranspiration datasets were reliable, as the downscaling procedure further improved the quality and spatial resolution of the CRU dataset and was concluded to be useful for investigations related to climate change across China [39]. The soil data were obtained from the National Tibetan Plateau Third Pole Environment Data Center constructed by the Food and Agriculture Organization of the United Nations (FAO) and the International Institute for Applied Systems (IIASA) in Vienna. The data source in China is the 1:1 million soil data provided by the Nanjing Soil Institute of the Second National Land Survey [40]. To overcome the effects of extreme values in a single year, we averaged the meteorological data over a five-year period for the interpolation in ArcGIS 10.4. As the study period was short, changes in soil properties organic carbon and root depth were assumed to be negligible. All data were converted into a unified projection coordinate system (Albers Conic Equal Area) and resampled to a spatial resolution of 300 m.

**Table 2.** Data descriptions and sources used in this study.

| Data Type | Data Requirements | Spatial Resolution | Data Source | Usage |
|---|---|---|---|---|
| Meteorological data | Monthly precipitation data from 1992 to 2003, 2008 to 2013, 2015 to2020 | 1 km | the National Earth System Science Data Center (http://www.geodata.cn (accessed on 21 March 2023)) | WY, WP, SR |
| | Monthly evapotranspiration data from 1992 to 2003, 2008 to 2013, 2015 to2020 | | | WY |
| Soil data | Root depth | 1 km | the Harmonized World Dataset ver1.2 of the National Tibetan Plateau Third Pole Enviornment Data Center (http://data.tpdc.ac.cn/ (accessed on 21 March 2023)) | WY |
| | Sand, silt, clay particles | | | WY, SR |
| | Organic carbon | | | WY, SR |
| | Weight capacity | | | WY |
| Satellite Image data | Digital elevation model | 30 m | NASA Earthdata Center (https://earthdata.nasa.gov/ (accessed on 21 March 2023)) | WP, SR |
| | NDVI data from 1992, 2000, 2010 and 2020 | 250 m | MOD13Q1 of Land Process Distributed Active Archive Centre (https://lpdaac.usgs.gov/ (accessed on 21 March 2023)) | WY, SR |

## 3. Results

### 3.1. CC and LUC from 1992–2020

Precipitation (Pre) and temperature (Tem) are the two major climate factors affecting HESs in northeast China. From 1992 to 2020, the overall climate of the northeast showed a trend of wetting and warming, with Pre increasing from 486.54 mm to 605.58 mm and Tem increasing from 3.26 °C to 3.72 °C (Figure 2). Among them, the most significant change of Pre was in the SJP, which increased from 546.35 mm to 774.40 mm; the most significant change of Tem was in the GKMR, which increased from 0.60 °C to 1.16 °C. From the perspective of each stage, Pre in the LRP fluctuated greatly, while Pre in the SJP continued to rise, and Tem in the rest of the regions showed a trend of "decreasing before increasing". In terms of spatial distribution, Pre in the LRP, northeastern GKMR and southern CBMR showed a trend of "decrease-increase-decrease", while Pre in the northern CBMR showed a trend of "increase-decrease-increase". Tem in the SJP showed a decreasing trend from 1992 to 2000, followed by a significant increase; in other regions, a decreasing trend from 1992 to 2010, followed by an increasing trend.

Affected by climate and human activities, cropland (+75,550.50 km$^2$) and built-up land (+13,136.76 km$^2$) expanded significantly in northeast China from 1992 to 2020, while grassland (−25,998.93 km$^2$) and wetland (−6923.87 km$^2$) shrank significantly. In particular, the area of cropland in the SNP, the SJP, and the LRP increased sharply, occupying a large amount of forest and grassland. From the perspective of spatial distribution, a large amount of forest and grassland in the central LKMR, the LRP, and the western CBMR was converted to cropland from 1990 to 2000; there was a significant increase in forest and grassland in the northern GKMR, western LKMR, and southeastern LRP, and frequent conversion of grassland and cropland in the southern GKMR, SNP, SJP, and LRP from 2000 to 2020. At the same time, built-up land occupied a large amount of cropland and grassland, especially in the SNP and SJP.

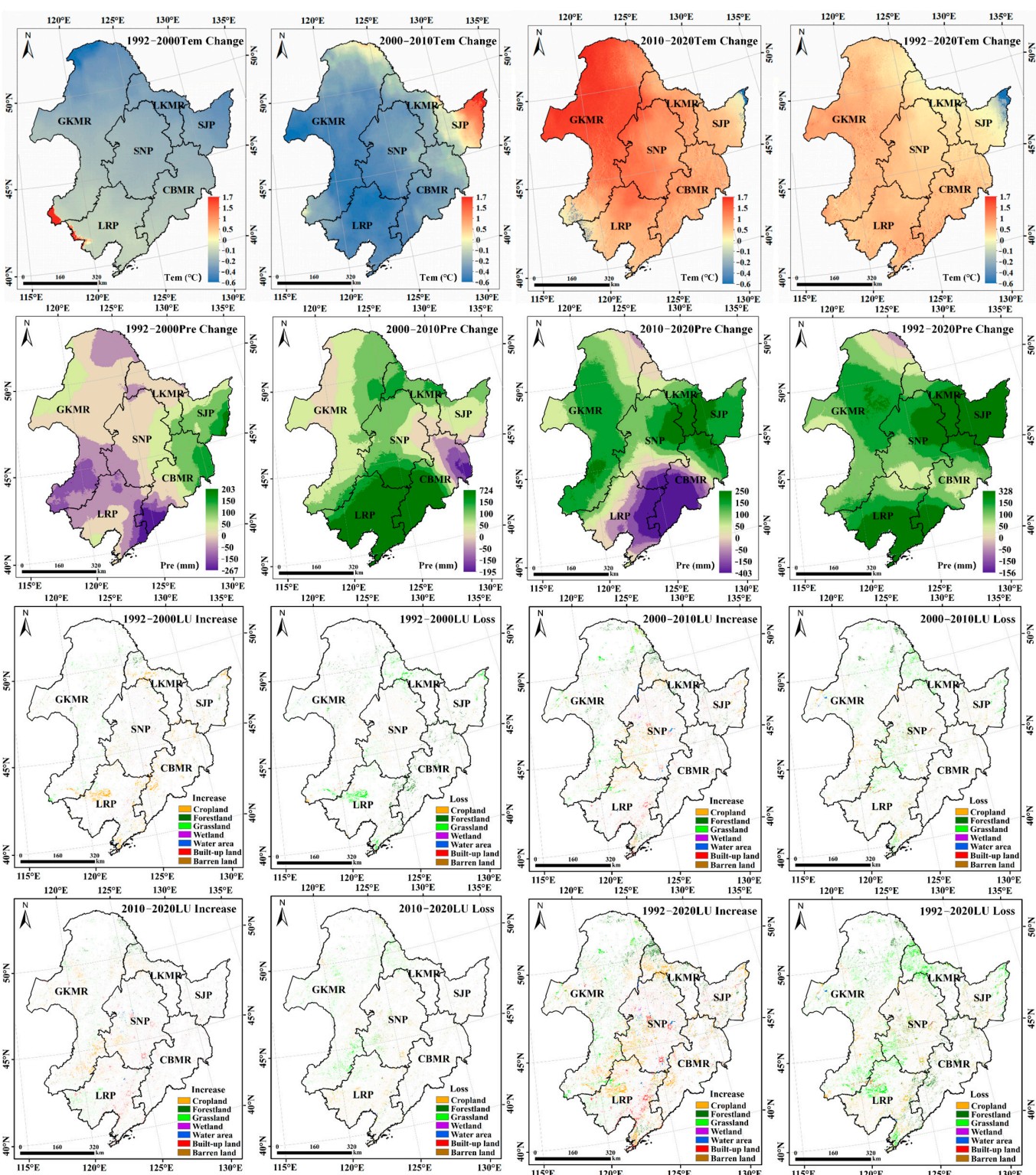

**Figure 2.** Spatial distribution of CC and LUC in Northeast China from 1992 to 2020.

### 3.2. HESs Change from 1992 to 2020

From 1992 to 2020, WY in northeast China increased from 130.48 mm to 224.81 mm and SR increased from $10.21 \times 10^3$ t/km$^2$ to $15.49 \times 10^3$ t/km$^2$. NE increased from 149.16 kg/km$^2$ to 224.06 kg/km$^2$, indicating a significant decrease in WP (Figure 3). Among them, WY, NE, and SR are all on the rise in all regions, with the largest increase in WY (+240.78 mm) and NE (+134.16 kg/km$^2$) in the SJP and the largest increase in SR (+7.24 $\times$

$10^3$ t/km$^2$) in the LKMR. From all stages, there was a continuous trend of increasing NE in the SNP, WY in the SJP, and SR in the, and the LKMR.

**Figure 3.** HESs change in Northeast China from 1992 to 2020.

There was significant spatial heterogeneity in WY, NE, and SR changes influenced by CC and LUC in northeast China from 1992 to 2020 (Figure 4). The spatial distribution of WY and Pre variation is roughly the same, with greater fluctuations in the LRP and CBMR. NE showed a significant increase in most regions, indicating a significant decrease in WP capacity in most regions, especially in the SJP, SNP, southern GKMR, and LRP from 1992 to 2020. The areas with a significant increase in SR were concentrated in the GKMR, LKMR, and CBMR. Among them, SR in the CBMR increased the most from 2000–2010, and SR in the GKMR and LKMR increased the most from 2010–2020.

### 3.3. Effects of CC and LUC on HESs from 1990 to 2020

3.3.1. Relative Effect of CC and LUC on HESs from 1990 to 2020

The change of WY was most influenced by CC, with CC having a greater impact on 77.61% of regional WY than LUC, with increased Pre causing an upward trend in WY across the region (Figure 5). The impact of CC on 94.38%, 65.41%, and 75.56% of regional WY was stronger than LUC for each of the phases 1992–2000, 2000–2010, and 2010–2020. WY in the SJP is most affected by climate change (1992–2000: 95.78%; 2000–2010: 46.21%; 2010–2020: 97.17%). In terms of spatial distribution, changes in WY in the eastern LRP, the southern CBMR, and the SJP are more influenced by climate change (Figure 6).

The change of NE was more affected by LUC, as LUC had a greater impact on NE than CC in 50.63% of the region from 1992–2020. In terms of the phases, CC had a greater impact on NE from 1992–2000 (67.62%), while the impact of LUC on WY was stronger than CC in subsequent years (2000–2010: 61.6%; 2010–2020: 72.55%). In particular, the impact of LUC on NE in the SNP was stronger than CC at all stages, with an increase in cropland area leading to a significant increase in NE. However, the increase in NE in the LRP from 2000 to 2020 was mainly influenced by CC. In terms of spatial distribution, change in NE in the western part of the LRP, the southern part of the GLMR and the southwestern part of the SNP was more influenced by LUC.

Both overall and regional SR change in the northeast was strongly influenced by LUC (northeast, 89.43%; CBMR, 78.57%; GKMR, 80.25%; LKMR, 84.02%; SNP, 99.42%; SJP, 90.10%; LRP, 87.23%). The impact of LUC on SR tended to increase across the different regions in terms of all phases. In terms of spatial distribution, the impact of LUC on SR in the GKMR was stronger than CC, while the impact of CC on SR in the CBMR and the LKMR was stronger than LUC and gradually increased in intensity.

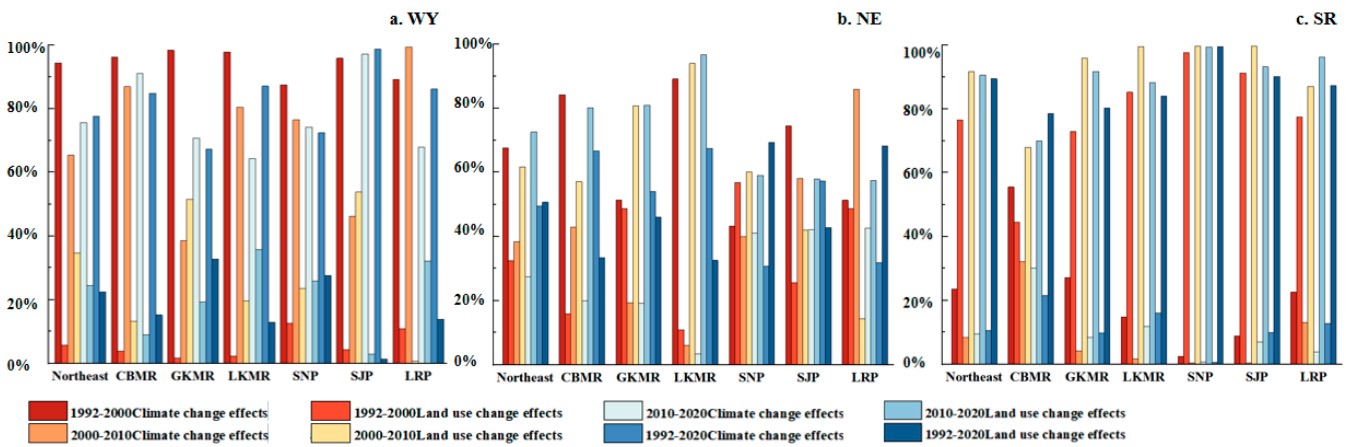

**Figure 4.** Spatial distribution of HESs change in northeast China from 1992 to 2020.

**Figure 5.** Percentage of pixels mainly influenced by climate and land use.

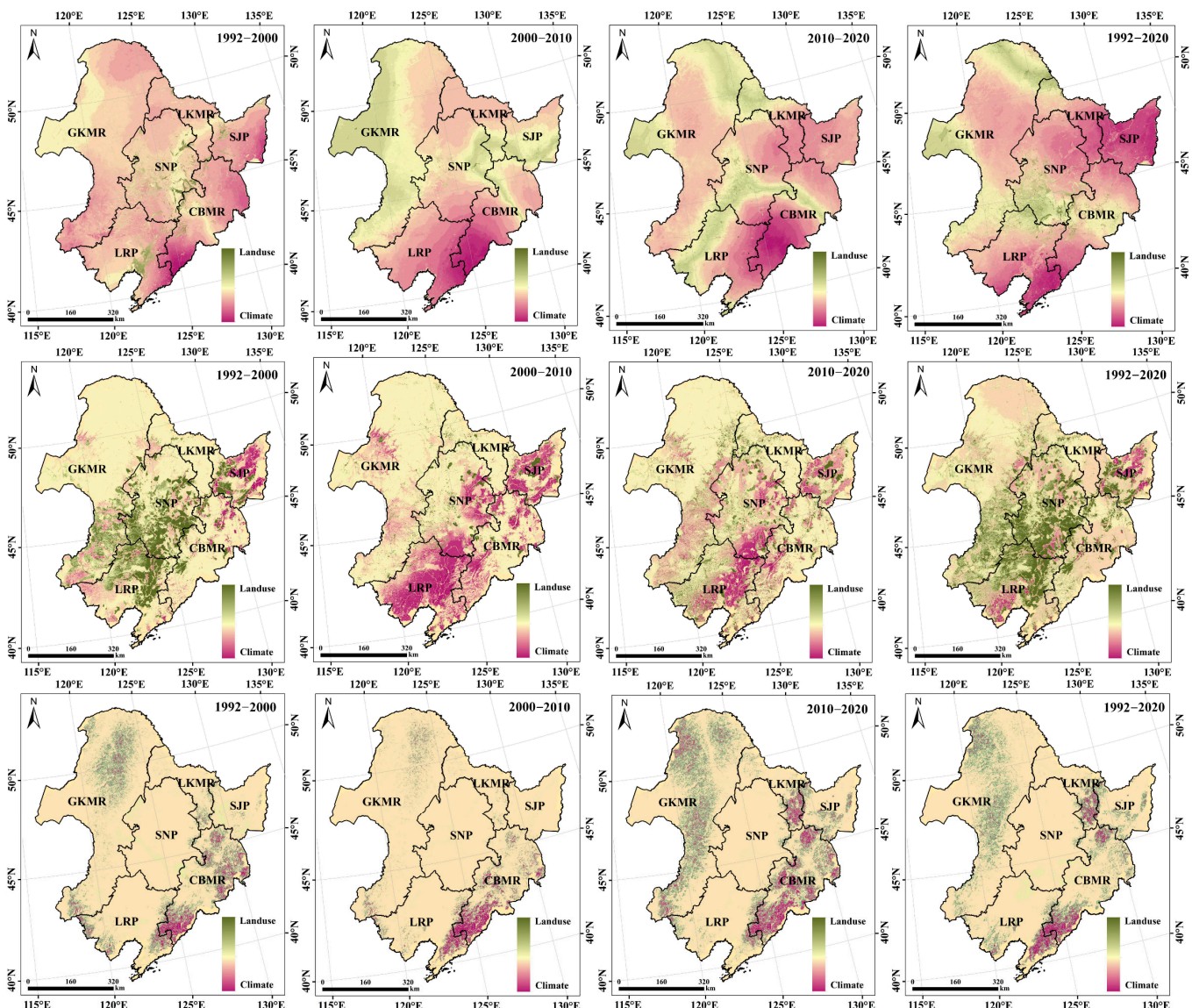

**Figure 6.** Relative importance of CC and LUC of HESs in Northeast China from 1992 to 2020.

### 3.3.2. Cumulative Effect of CC and LUC on HESs from 1990 to 2020

CC and LUC interaction had a inhibitory effect on WY in 9.74% of the region, NE in 24.68% of the region, and SR in 4.21% of the region (Figure 7). Among them, the interaction between CC and LUC had the highest inhibitory effect on WY in the GKMR and LRP (18.84%), the highest inhibitory effect on NE in the SNP (37.40%), and the highest inhibitory effect on SR in the CBMR (8.01%) (Figure 8). The contribution of CC and LUC interaction to HESs was at a maximum in all regions from 2000–2010.

Although CC and LUC contributed to all HESs in the northeast at all stages, it did so to a lesser extent for the vast majority of the region. In terms of spatial distribution, the interaction between CC and LUC had a higher inhibitory effect on WY and NE in the southwestern SNP, central LRP and southern GKMR, and a higher degree of inhibition on SR in the CKMR, eastern LRP and southern GKMR. The interaction between CC and LUC contributed more to NE in the central part of the SJP and LRP and more to SR in the northern part of the GKMR.

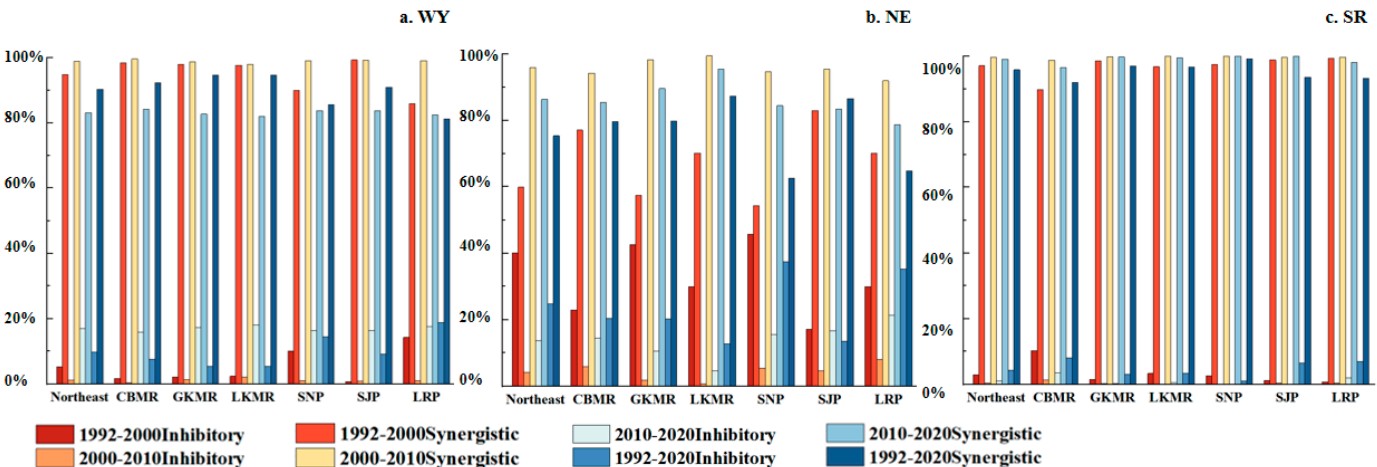

**Figure 7.** Percentage of pixels mainly influenced by inhibitory and synergistic.

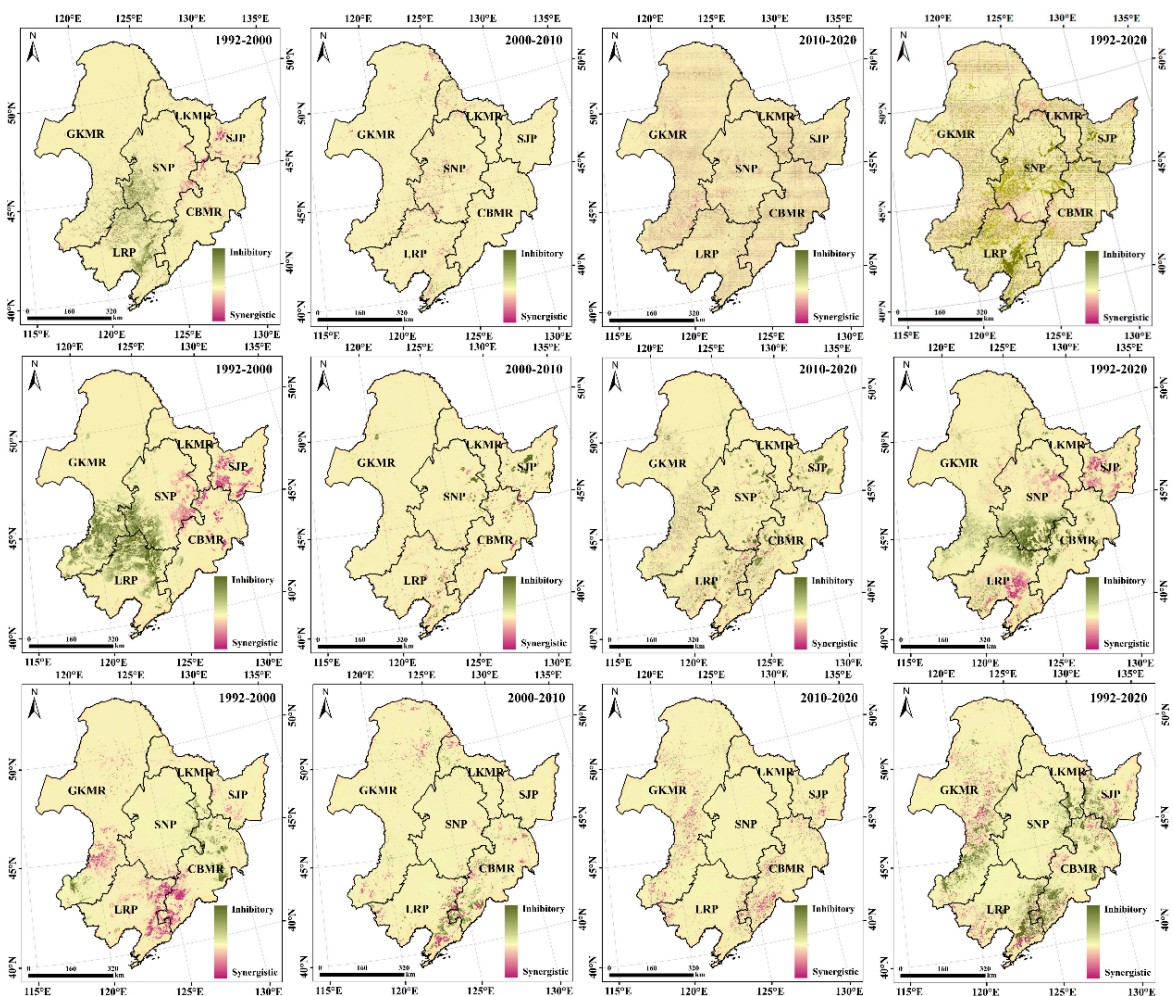

**Figure 8.** Combined effects of CC and LUC of HESs in northeast China from 1992 to 2020.

## 4. Discussion

### 4.1. Effects of CC and LUC on HESs

The spatial heterogeneity of climate and land use pattern change driven by policy in northeast China is an important reason for the change of HESs [26]. This study combines scenario analysis with ecosystem services assessment to provide an effective method for analyzing the relative importance (RII) and joint impact (CEI) of climate change and land

use change on ecosystem services [41]. Our findings suggest that the independent impacts of climate change and land use change and their interactions place significant pressure on the ability of ecosystems to provide hydrological ecosystem services

Precipitation directly influenced regional water input in northeast China (thus affect surface hydrological processes) [34]. WY increased with precipitation in most regions of the northeast, and CC had a greater impact on WY than LUC, especially in the LRP and the SNP. However, with the implementation of ecological restoration projects such as returning farmland to forest and natural forest protection projects, forestland and grassland has expanded in a large area, especially in the GKMR (Figure 9) [26]. Because the gain effect of forestland increase was smaller than the water loss caused by evapotranspiration, the water yield decreased instead of increased [42,43]. A recent study has indicated that warming in the northeast increased potential evapotranspiration by 6.15% from 1990 to 2020, reducing the moisture-holding capacity of the atmosphere and exacerbating drying conditions [44]. Therefore, climate warming and ecological land expansion could promote the increase in evapotranspiration rate in the GKMR, but inhibit the increase in water yield.

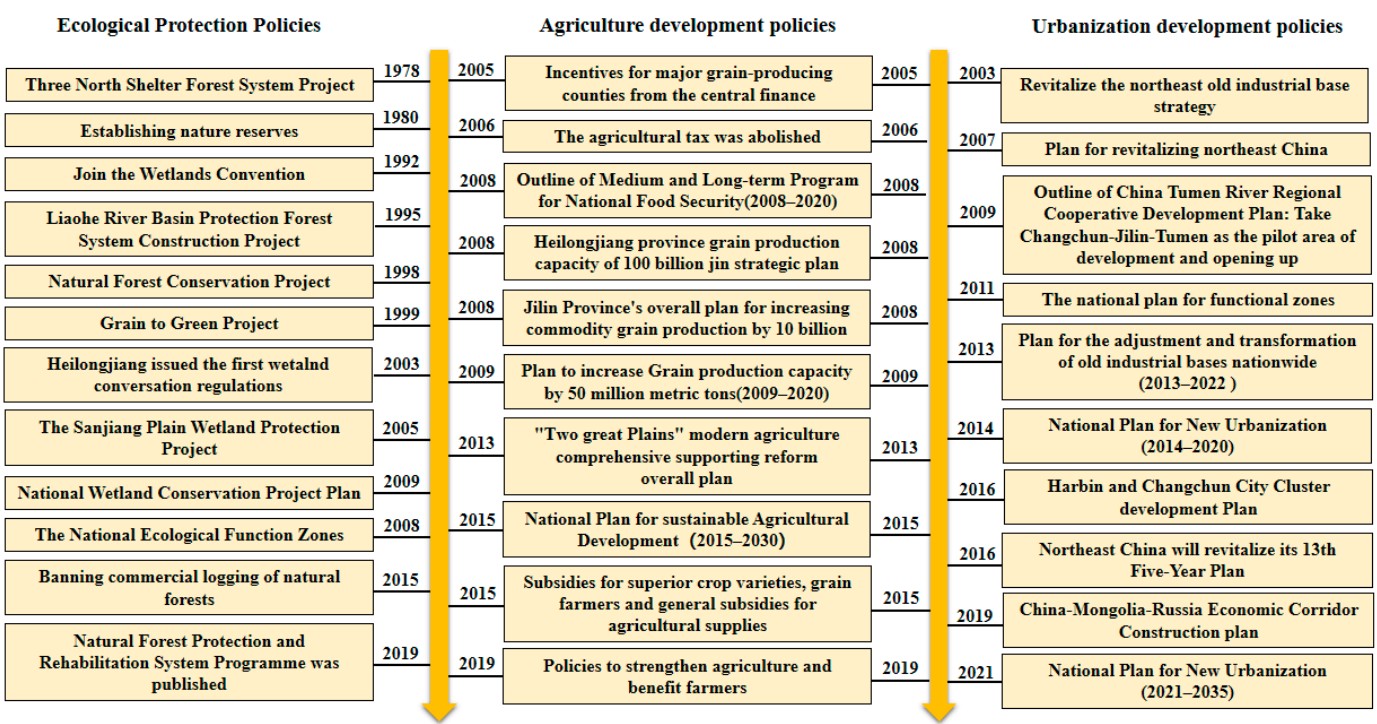

**Figure 9.** A summary of policies in northeast China for ecological protection, agricultural development, and urbanization development.

Driven by a series of agricultural development policies such as the "SNP and the SJP as the core of the grain production capacity building project", and urbanization development policies such as the "revitalization of the old industrial base strategy in the Northeast", large areas of wetland and grassland have been reclaimed for cropland and built-up land, reducing the ability to intercept and filter nitrogen and reducing the water purification capacity [45,46]. Under the background of the expansion of cropland and built-up land, due to the needs of agricultural production and human life, fertilizers and pesticides, irrigation sewage and production and domestic waste have increased significantly, resulting in a significant increase in NE in the SJP, the SNP, the LRP, and areas with concentrated built-up land due to the concentration of cropland and high irrigation demand, seriously affecting shallow groundwater quality [35]. Thus, LUC has contributed significantly to the increase in NE. In addition, the increase in precipitation has accelerated nitrogen flow through the surface and underground, which has further promoted the increase in nitrogen export in the SNP and LRP [47].

Due to the temperate monsoonal continental climate in northeast China, there is no consistent heavy precipitation throughout the year [48]. Therefore, CC had a relatively small impact on SR, while LUC had a significant impact on SR. The expansion of the forestland significantly enhanced the ability of the root soil retention and sediment retention ability, thus improving SR, especially in the northern GKMR, the southern CBMR and the eastern LRP, where the afforestation project was implemented [49]. In recent years, the precipitation of the SJP and the SNP has increased significantly, with a consequent increase in rainfall erosion, which, together with the reduction in the area of forestland and grassland with high vegetation cover, further reduced the soil retention ability [50].

### 4.2. Strategies and Implications

Our results show that CC and LUC, driven by multiple land-use decisions in the northeast significantly improved SR and reduced WY in mountainous areas. At the same time, LUC significantly influenced WP and SR. The finding highlights the importance of integrating policy-driven LUC and its interaction with CC on HESs into management practices and land-use decisions in order to achieve sustainability of ecosystem services.

First, we should steadily promote key forestry projects such as Natural Forest Protection, establish a forest resource ecological protection system, and strictly limit destructive construction activities. In addition, land use structure, tree species structure, and stand density should be adjusted to improve the quality of forestland ecosystems, thereby mitigating the increase in evapotranspiration caused by forestland expansion, in order to reduce the impact of the combined effect of CC and LUC on WY [36]. It has been shown that increasing canopy density can effectively reduce evaporation by reducing effective radiation and wind speed [34]. Secondly, in the southern part of the GKMR and the western part of the LRP, there is an urgent need to implement comprehensive grassland management projects, sealing projects, and rotational grazing projects to improve grassland protection and thus control the conversion of cropland and grassland. The southwestern part of the SNP is in urgent need of a virtuous cycle of wetland protection and rational utilization through the implementation of wetland restoration projects, strengthening the management of wetland nature reserves and improving the accountability mechanism for wetland protection. Through the protection of forestland, grassland and wetland, the ability of nitrogen interception and filtration and sediment interception can be increased.

Blindly reclaiming cropland at the expense of ecological land is not a reasonable way to increase food production. It is necessary to balance food production function and ecosystem service function. The SJP, the central LRP, and the eastern part of the SNP should make use of the long-term feedback mechanism for the conversion of cropland and ecological land to establish a flexible transformation space; rationally plan the main food producing areas, ecological function areas, and flexible conversion areas; and promote the integration of cropland protection and ecological protection policies. At the same time, artificial ecosystems can be created by constructing cropland shelterbelts to slow down soil erosion, improve climate and hydrological conditions, and effectively block nitrogen losses caused by fertilization [51,52]. Therefore, there is an urgent need to implement an integrated development strategy including mountains, water, forests, fields, lakes, grasses, and sands in northeast China, to build a security model for the coordinated development of multiple ecosystems, and to bring into play the ecological protection of ecological land for cropland and improve the effectiveness of implementing ecological protection and restoration projects.

### 4.3. Limitations

This study provides some clues about the impacts of CC and LUC on HESs and provides a scientific basis for formulating reasonable land use management and ecosystem policies. However, there are still some issues remaining that deserve further attention: (1) Although the InVEST model is a suitable tool to reflect changes in HESs at multiple scales, the InVEST model does not take into account certain hydrological processes (e.g., surface

runoff, surface runoff, and soil runoff) and interactions between surface and groundwater. At the same time, daily or monthly extreme weather events are ignored. In the future, the model principle improvement, parameter verification and other aspects will be further studied, and field observation work will be increased to obtain measured data to support the research results. (2) Although RII and CEI can be used to assess the impacts of CC and LUC on changes in HESs, only two factors or groups of factors can be considered at this stage of the analysis and the mechanisms of interaction between CC, LUC, and changes in HESs cannot be clarified. In the future, the mechanisms of ecosystem service change under various climate factors and land-use type transformations will be analyzed more carefully in order to generate more effective ecological management policies and recommendations. (3) This study examines the effects of climate change and land use change on changes in HESs in 1992, 2000, 2010 and 2020, which to some extent ignores the interannual dynamics from 1992–2020. Since most climatic factors have distinct seasonal characteristics, with lagging effects on ecosystem services, time scales should also be considered. In the future, long-term annual data series will be applied for comprehensive analysis to identify the mechanisms by which climate and land-use change affect changes in HESs at different time scales.

## 5. Conclusions

This study analyzed CC and LUC driven by land-use policies in the northeast from 1992 to 2020. Using climate and land use data for four periods (1992, 2000, 2010, and 2020), four scenarios were constructed. The relative and cumulative effects of CC and LUC on HESs were quantified through the Relative Importance Index (RII) and the Combined Effects Index (CEI). The results show that the overall climate in northeast China was wetter and warmer (Pre + 119.04 mm, Tem + 0.46 °C), and the LUC was "a sharp increase in cropland and built-up land, and a significant shrinkage in grassland and wetland" from 1992 to 2020. WY, NE, and SR in the northeast were all on an upward trend. Due to the large spatial variation in CC and LUC in the northeast, there was significant spatial heterogeneity in WY, NE, and SR changes. CC had a much greater impact on WY than LUC, particularly in the eastern SJP, the eastern LRP, and the southern CBMR, where the trend of rising precipitation is more pronounced. NE and SR were strongly influenced by LUC. With the conversion of a large amount of forest and grassland to cropland and built-up land in the western LRP, southern GKMR, and southwestern SNP, NE rose significantly. As the forestland expands, SR rose extremely significantly. It is worth noting that the interaction between CC and LUC had a catalytic effect on HESs in most regions of the northeast, but was inhibitory in some regions. The interaction of climate warming and forestland expansion in the northern GKMR resulted in a significant increase in potential evapotranspiration and a dramatic decrease in WY.

Despite the limitations of this study, the findings have practical, methodological and policy-related implications to support the application of HESs in land use planning and to develop more effective ecosystem conservation policies. Our study can also help policy-makers to develop more comprehensive, spatially adapted climate-resilient management assumptions to promote sustainable ecosystem service provision.

**Author Contributions:** Conceptualization, methodology, software, visualization, writing—original draft preparation, M.W.; writing—review and editing, supervision, funding acquisition, G.L. All authors have read and agreed to the published version of the manuscript.

**Funding:** This work was supported by the National Natural Science Foundation of China (Grant No. 41671520).

**Data Availability Statement:** The data presented in this study are available on request from the corresponding author.

**Conflicts of Interest:** The authors declare no conflict of interest.

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
