# Peer review of "Relative and Cumulative Effects of Climate and Land Use Change on Hydrological Ecosystem Services in Northeast China"

_land, doi:10.3390/land12071298_

Round 1

Reviewer 1 Report

This study quantified the spatio-temporal change of HESs (water yield, water purification, soil retention) from 1992 to 2020 in the Northeast China, and evaluated the relative contribution and cumulative effects of CC and LUC on HESs through environmental setting scenarios and using two indicators (the Relative Importance Index and the Combined Effects Index). 

This study has done a lot of work, but it is recommended that the author make the following modifications.

introduction: "Lack of assessment of the impacts of CC and LUC on HESs at

large spatial scales where there is significant environmental heterogeneity, thereby reduc-

ing the regional utility of ecosystem management responses. Therefore, there is an urgent

need to study the relative and cumulative effects of LUC and CC on HESs."

If this is a research gap, more clues must be provided.

Data requirement and preparation:Please provide the year of the data.

Discussion:The discussion section should be a response to the research question. At present, this part is still the research result.

Extensive editing of English language required

Author Response

Point 1: introduction: "Lack of assessment of the impacts of CC and LUC on HESs at large spatial scales where there is significant environmental heterogeneity, thereby reducing the regional utility of ecosystem management responses. Therefore, there is an urgent need to study the relative and cumulative effects of LUC and CC on HESs." If this is a research gap, more clues must be provided.

Response 1: Accept. Thanks for your advice. At present, most studies have focused on administrative, watershed, typical regional and ecological reserve scales, or analysed the impact of LUCC on HESs as a result of a single decision. Although studies have also studied the effects of climate change and land use change on changes in hydrological ecosystem services at large spatial scales, studies at large spatial scales have mainly focused on the same type of area, with no significant spatial heterogeneity in topography, vegetation or climatic characteristics. For specific changes see line number 77-83.

Point 2: Data requirement and preparation:Please provide the year of the data.

Response 2: Accept. Thanks for your advice. To overcome the effects of extreme values in a single year, we averaged the meteorological data over a five-year period for the interpolation in ArcGIS 10.4. I have relabelled the data year: Monthly precipitation data from 1992 to 2003, 2008 to 2013, 2015 to2020; Monthly evapotranspiration data from 1992 to 2003, 2008 to 2013, 2015 to2020; NDVI data from 1992, 2000,2010 and 2020. For specific changes see line number 243.

Point 3: Discussion:The discussion section should be a response to the research question. At present, this part is still the research result.

Response 3: Accept. Thanks for your advice. The discussion section of this study has explained how climate change, policy-driven land-use change and their interactions affect hydrological ecosystem services. There may have been confusion in the logic during the interpretation, which has been further modified. For specific changes see section 4.1. At the same time, we put forward corresponding suggestions and measures on how to improve the hydrological system services. For specific changes see section 4.2.

Point 4: Extensive editing of English language required.

Response 4: Accept. Thanks for your advice. I have made corrections to the English language.

Reviewer 2 Report

Dear authors.

I am grateful for the opportunity to review the article "Relative and cumulative effects of climate and land use change on hydrological ecosystem services in Northeast China

". The article is devoted to an important and relevant topic of the relationship between climate change and land use/land cover.

The article has a large number of advantages (structured manuscript, rich illustrative material, etc.) which I don't see the point in listing. But the article has a few comments. Eliminating comments will help improve the quality of the article. The article can be accepted after the elimination of comments.

1. I recommend the authors to proofread the manuscript of the article more carefully. There are inaccuracies in the design (for example, the list of references, the design of drawings, typos, etc.).

2. In Figure 1, put the neighboring countries of China. Sign large geographical objects – names of countries, seas, oceans. The relief map does not indicate the units of change (meters or feet?)

3. In Figure 2, there are no units of change in the legends of geographical maps (maps in 1 and 2 lines / horizontal line). A similar problem is shown in Figure 4.

4. Figures 3,4, 7, 9 are poorly readable. Increase their size or captions on the drawings.

5. In section 2, explain more clearly why the data you provided was selected? What are their advantages compared to similar data from other sources (Table 2)?

6. I recommended that you remove the numbering of the results from the annotation.

Author Response

Point 1: I recommend the authors to proofread the manuscript of the article more carefully. There are inaccuracies in the design (for example, the list of references, the design of drawings, typos, etc.)

Response 1: Accept. Thanks for your advice. I have re-proofed the entire text and carefully revised the references, drawing and typos, etc.

Point 2: In Figure 1, put the neighboring countries of China. Sign large geographical objects – names of countries, seas, oceans. The relief map does not indicate the units of change (meters or feet?)

Response 2: Accept. Thanks for your advice. I have made changes to Figure 1. I put the neighboring countries of China, sign large geographical objects, names of countries, seas, oceans. At the same time. For specific changes see line number 139.

Point 3: In Figure 2, there are no units of change in the legends of geographical maps (maps in 1 and 2 lines / horizontal line). A similar problem is shown in Figure 4.

Response 3: Accept. Thanks for your advice. I have readded change units to the legend of the full-text diagram.

Point 4: Figures 3,4, 7, 9 are poorly readable. Increase their size or captions on the drawings.

Response 4: Accept. Thanks for your advice. I've been resized to increase readability.

Point 5: In section 2, explain more clearly why the data you provided was selected? What are their advantages compared to similar data from other sources (Table 2)?

Response 5: Accept. Thanks for your advice. I have explained the data sources and their advantages compared to other data sources. The land use were provided by the Climate Change Initiative (CCI, http://maps.elie.ucl.ac.be/CCI/viewer). The relatively high spatial resolution of the data and its long-term consistency, annual updates and high thematic detail on a global scale make it attractive for numerous applications such as land accounting, forest monitoring, and scientific research. Dominant land use categories agriculture and forest show an agreement of over 80 %. The monthly precipation and evapotranspiration dataset were reliable, as the downscaling procedure further improved the quality and spatial resolution of the CRU dataset and was concluded to be useful for investigations related to climate change across China. The soil data were obtained from the National Tibetan Plateau Third Pole Enviornment Data Center constructed by the Food and Agriculture Organization of the United Nations (FAO) and the International Institute for Applied Systems (IIASA) in Vienna. The data source in China is the 1:1 million soil data provided by the Nanjing Soil Institute of the Second National Land Survey . For specific changes see line number 222-237.

Point 6:I recommended that you remove the numbering of the results from the annotation.

Response 6: Accept. Thanks for your advice. I have removed the numbering of the results from the annotation

Round 2

Reviewer 1 Report

Well done.